# Thrombotic Events Are Unusual Toxicities of Chimeric Antigen Receptor T-Cell Therapies

**DOI:** 10.3390/ijms24098349

**Published:** 2023-05-06

**Authors:** Christopher Schorr, Jorge Forindez, Manuel Espinoza-Gutarra, Rakesh Mehta, Natalie Grover, Fabiana Perna

**Affiliations:** 1Department of Medicine, Indiana University School of Medicine, Indianapolis, IN 46202, USA; cschorr@iu.edu (C.S.);; 2Department of Biomedical Engineering, Purdue University, West Lafayette, IN 47907, USA; 3Department of Medicine, University of North Carolina School of Medicine, Chapel Hill, NC 27599, USA; 4O’Neal Comprehensive Cancer Center, University of Alabama, Birmingham, AL 35233, USA

**Keywords:** chimeric antigen receptor (CAR) T-cell, thrombotic events, cytokine release syndrome (CRS), immune effector cell-associated neurotoxicity syndrome (ICANS), hematological malignancies, disseminated intravascular coagulation (DIC), toxicity, multiple myeloma, non-Hodgkin lymphoma (NHL)

## Abstract

Chimeric antigen receptor (CAR) T-cell therapy has greatly transformed the treatment and prognosis of B-cell hematological malignancies. As CAR T-cell therapy continues to be more readily adopted and indications increase, the field’s recognition of emerging toxicities will continue to grow. Among the adverse events associated with CAR T-cell therapy, cytokine release syndrome (CRS) and immune effector cell-associated neurotoxicity (ICANS) are the most common toxicities, while thrombotic events represent an under-reported, life-endangering complication. To determine thrombosis incidence post CAR T-cell therapy, we performed a multi-center, retrospective study on CAR T-cell therapy adult patients (N = 140) from Indiana University Simon Cancer Center and the University of North Carolina Medical Center treated from 2017 to 2022 for relapsed and refractory B-cell acute lymphoblastic leukemia (B-ALL, N = 3), diffuse large B-cell lymphoma (DLBCL, N = 92), follicular lymphoma (FL, N = 9), mantle cell lymphoma (MCL, N = 2), and multiple myeloma (MM, N = 34). We report 10 (7.14%) thrombotic events related to CAR T-cell therapy (DLBCL: N = 8, FL: N = 1, MM: N = 1) including 9 primary venous events and 1 arterial event that occurred with median time of 23.5 days post CAR T-cell infusion. In search of parameters associated with such events, we performed multivariate analyses of coagulation parameters (i.e., PT, PTT, and D-Dimer), scoring for adverse events (Padua Score and ISTH DIC Score) and grading for CAR T-cell toxicity severity (CRS grade and ICANS grade) and found that D-Dimer peak elevation and ICANS grade were significantly associated with post-CAR T-cell infusion thrombosis. While the pathophysiology of CAR T-cell associated coagulopathy remains unknown, our study serves to develop awareness of these emerging and unusual complications.

## 1. Introduction

Chimeric antigen receptor (CAR) T-cell therapy has revolutionized the treatment of B-cell hematological malignancies and continues to show great promise for other malignancies. Since 2017, several FDA-approved CAR products have become available (idecabtagene vicleucel, liscoabtagene maraleucel, tisagenlecleucel, brexucabtagene autoleucel, and axicabtagene ciloleucel) with many others currently under clinical investigation. CAR T-cell therapy is approved for relapsed and refractory (r/r) B-cell acute lymphoblastic leukemia (B-ALL), diffuse large B-cell lymphoma (DLBCL), follicular lymphoma (FL), mantle cell lymphoma (MCL), and multiple myeloma (MM).

While effective, CAR T-cell therapy commonly presents with adverse events such as cytokine release syndrome (CRS) and immune effector cell-associated neurotoxicity syndrome (ICANS) [1]. The systemic inflammatory response in CRS and ICANS involves a massive release of multiple cytokines such as interleukin IL-1, IL-6, IL-10, interferon gamma (IFN-γ), and tumor necrosis factor alpha (TNF-α) [2]. As of December 2022, the FAERS database reports 7510 patients treated with approved CAR T-cell products have presented with an adverse event. Of these patients, the incidence of toxic effects associated with death was 4.3% (325 patients). The 2 most common toxicities reported are CRS and ICANS occurring in 50% (3758 patients) and 15% (1134 patients), respectively. 

With the rapidly growing list of indications, the increasing number of CAR T-cell cellular targets, and thus, their evolving toxicity profiles, there is a crucial need to develop awareness of the emerging and atypical adverse events that extend beyond the current paradigm of CRS and ICANS. For example, recent studies and case reports have described rare complications following CAR T-cell therapy, such as unusual neurologic and immune reactions [3,4,5]. Disseminated intravascular coagulation (DIC) is a coagulation complication associated with severe CRS [6,7]. Incidences of laboratory DIC related to CAR T-cell therapy range between 7% (*n* = 7) and 28.3% (*n* = 15) according to 2 studies in 100 (B-ALL, DLBCL, and MM) and 53 (B-ALL) patient cohorts, respectively [8,9]. These findings suggest the need for the awareness, monitoring, and management of coagulation disorders in patients undergoing CAR T-cell therapy [10]. 

Very little is known regarding the association between thrombotic risk and CAR T-cell therapies. Of the studies published so far, Johnsrud et al. observe an incidence of 6.3% (*n* = 8) thrombosis within 3 months in their 127 DLBCL or B-ALL patient cohort treated with axicabtagene ciloleucel or bispecific CD19/CD22 CAR T-cells [11]. Melody et al. report that 1 DLBCL patient out of 103 DLBCL or FL patients receiving CD-19 CAR T-cell therapy experienced a central-line-associated thrombosis [12]. Parks et al. note incidences of venous thrombosis embolism (VTE) in 11% (*n* = 4) and 7% (*n* = 4) in 37 DLBCL and 54 MM patients, respectively, within 60 days post-CD19 CAR T-cell infusion [13]. Lastly, Hashmi et al. report that 11% (*n* = 16) of 148 DLBCL patients experienced a new episode of VTE between day 0 and day 100 after CD-19 CAR T-cell infusion [14]. The Wilson score interval of these studies is 9.5% thrombotic event occurrence rate of total patients with a 95% confidence interval of 2.6%. 

Using a large and diverse cohort of patients, we first performed a retrospective analysis of 140 patients with r/r B-ALL, DLBCL, FL, MCL, and MM treated with CAR T-cell therapies currently approved by the FDA at 2 institutions. Furthermore, we determined whether there was any conventional risk factor or concurrent CRS- or ICANS-promoted development of these complications. 

## 2. Results

We retrospectively collected data from the electronic medical record (EMRs) of 140 patients with CAR T-cell treated r/r B-ALL (*n* = 3), DLBCL (*n* = 92), FL (*n* = 9), MCL (*n* = 2), and MM (*n* = 34). We found that of the 140 patients, 7.1% (*n* = 10) developed a complication with thrombotic clinical presentations (Table 1) post-CAR T-cell treatment. The primary thrombotic events included four pulmonary embolisms (PEs), four deep venous thromboses (DVTs), one coronary artery thrombosis resulting in non-ST elevation myocardial infarction (NSTEMI), and one cerebral vein sinus thrombosis (CVST). In total there were nine venous complications (DVT, PE, and CVST) and one arterial (NSTEMI). All patients with thrombosis had a successful resolution except for the patient who developed a chronic CVST. One patient had a central-line-associated thrombosis. All DVTs were treated with direct oral factor X inhibitors (either apixaban or rivaroxaban) and PEs were treated with either heparin, LMWH, or factor X inhibitors. 

All eight DLBCL patients and the FL patient with thrombosis received axicabtagene ciloleucel. The MM patient with thrombosis received ciltacabtagene autoleucel. The CAR-T treatment time to primary thrombotic event varied greatly with a range between 3 and 346 days. The PE events occurred on days 13, 30, 41, 46, and 346 post-CAR T-cell infusion. The DVT events occurred on days 5, 10, 17, and 113 post-CAR T-cell infusion. A DLBCL patient (patient #3) with recurrent DLBCL retroperitoneal adenopathy developed a CVST on day 15 post-CAR T-cell infusion with the concurrent discovery of a retroperitoneal hematoma made by abdominal CT. Anticoagulation therapy was not administered and the retroperitoneal hematoma was managed appropriately. Approximately 6 months later, this patient developed bilateral PEs and a DVT in their right calf vein at days 218 and 219, respectively. Interestingly, we report a coronary artery thrombosis with NSTEMI in a DLBCL patient treated with CAR T-cell therapy. This patient had a previous medical history of coronary artery disease and myocardial infarction with a prior right leg DVT five months before CAR T-cell infusion. Chest pain and elevated troponin were present without significant EKG changes three days after CAR T-cell infusion. Percutaneous coronary intervention (PCI) was performed to successfully resolve the thrombosis. No thrombosis recurrence was noted on patients who underwent anticoagulation treatment. 

To investigate the potential baseline (pre-CAR T-cell therapy) factors that could correlate with the development of these toxicities, we stratified pre-CAR T infusion patient characteristics with thrombosis events (Table 2). We found no statistically significant associations between thrombotic and non-thrombotic patients regarding patient age (median: 60 years, range: 17–81 years), sex, diagnosis, BMI (median: 27, range: 17.6–53.7), prior history of bleeding, venous thromboembolism or pre-existing cardiovascular condition, and anticoagulation prophylaxis (preceding CAR T-cell infusion), suggesting the existence of CAR-T-cell-therapy-related factors beyond the conventional risk factors for thrombosis in the general population. Notably, most patients on anticoagulation therapy prior to CAR-T cell therapy had history of a thrombotic condition. In addition, we analyzed the number of prior lines of therapy patients received before CAR T-cell therapy. Patients received a median of 3 prior lines of therapy (range = 1–15) including prior autologous transplant in 50 (35.7% [*n* = 18 DLBCL, *n* = 30 MM, *n* = 2 FL]). A total of 48 (34.2%) had additional bridging therapy and 18 patients (12.9%) received anticoagulation medication prior to CAR T-cell infusion. Elevated lactate dehydrogenase (LDH), which serves as an indirect measure of disease burden in lymphomas, was measured to exclusively compare patients with lymphomatous disease (ALL and DLBCL); there was no statistically significant difference in LDH elevation status of thrombotic and non-thrombotic lymphoma patients. 

To further investigate the relationship between thrombosis events and CAR T-cell therapy we analyzed the association between thrombosis, several coagulation parameters, and CRS/ICANS grade post-CAR T-cell infusion (Table 3). We found that a D-dimer peak elevation of three times the upper normal limit (UNL) was most associated with a thrombotic event (*p*-value = 0.03), suggesting that D-dimer peak levels may be a sensitive indicator to predict patients at risk of developing thrombosis post-CAR T-cell infusion. PT, aPTT, fibrinogen, platelet nadir, and ferritin peak differences were statistically insignificant between thrombotic and non-thrombotic groups. Similarly, the baseline platelet laboratory value difference between thrombotic and non-thrombotic patients was not statistically significant. Predictive scores (i.e., ISTH DIC score and Padua prediction score) were calculated for each patient and there were no significant associations.

All patients were evaluated for ICANS and CRS throughout their 30 day hospitalization period. We found that the presence of ICANS (grade ≥ 1) was significantly associated with the occurrence of thrombotic events (*p* = 0.04). Conversely, there was no significant association between the presence of CRS and occurrence of thrombotic events (*p* > 0.99). When analyzed based on severity (a comparison between low (grades 0–1) and high (grades 3–4)), there were no statistically significant associations between high-grade ICANS (*p* = 0.43) and CRS (*p* = 0.37) with thrombotic events; this suggests that the presence of ICANS (grade ≥ 1) may determine thrombosis risk. 

## 3. Discussion

The toxicity profile of CAR T-cell therapy is rapidly evolving as the number of treated patients continues to increase, novel products become FDA-approved, and clinical trials with new targets and indications open [15,16]. A better understanding of the risks of CAR T-cell therapy may help clinical management [17,18,19]. We report an incidence of thrombotic complications of 7.1% in 140 patients, similarly to the study by Johnsrud et al. reporting that 6.3% (*n* = 8 out of 96) of patients undergoing CAR T-cell therapy develop thrombosis [11]. Both of our studies agree on the significant association between ICANS grade, but not CRS grade. We also found that this association, existing in most if not all CAR T-cell treated patients, may include arterial events. It is notable that, in our cohort, the majority of thrombotic events occurred in patients who received treatment for B cell lymphomas with an incidence of 9.7% compared to an only 2.9% incidence in patients being treated for MM with, in the existing literature, VTE incidence in B cell lymphomas ranging from 1.4 to 4.2% and in MM from 5 to 12% [20]; it is possible that the higher rates of VTE in lymphoma patients is driven by the higher rates of ICANS in lymphoma-directed products when compared to MM-directed products [21,22], which reinforces the theory that systemic inflammation plays an important role in VTE development. The association between ICANS and VTE development, but not CRS, is unclear and it may be due to differences in cytokine profiles or temporal differences in their presentation after CAR T-cell infusion [23]. 

We observe that the D-dimer peak and the presence of ICANS are significantly elevated in thrombotic patients after infusion of CAR T-cells and thus may serve as useful parameters to predict thrombosis risk. We further report that the time to primary thrombotic event can vary greatly with an average duration of 63 days post-CAR T-cell infusion. Thrombotic events that occur earlier (within 30 days post-CAR T-cell infusion) may have a different etiology than the long-term thrombotic events that occur outside of frequent hospital monitoring. We recognize that the patient who developed thrombosis at day 346 post-CAR T-cell infusion may be an outlier. When adjusted for short-term events, 6 patients experienced a thrombotic event within an average of 13 days post-CAR T-cell infusion. While the pathophysiology of these toxicities remains unknown, it is possible that an increased risk of thrombosis is due to endothelial cell activation and vascular injury [24]. Several groups have shown that high-grade ICANS can generally present with an increased von Willebrand factor, ratios of ANG2: angiopoietin 1, and an endothelial activation and fibrin deposition [11]. These factors may contribute to and further worsen the hypercoagulable state of patients undergoing CAR T-cell treatment especially considering that patients undergoing stem cell transplant have a risk of developing a VTE of 1–4% [25]. Therefore, our study emphasizes the need for thoroughly monitoring patients undergoing CAR T-cell therapy and optimizing their coagulation prophylaxis and treatment.

## 4. Materials and Methods

Mini Systematic Review: To better understand the occurrence of thrombotic events related to CAR T-cell therapy in the published literature, we first conducted a mini systematic review of the available studies. We searched PubMed, Embase, and Web of Science databases for articles reporting thrombotic events in patients receiving CAR T-cell therapy. Relevant articles were identified using a combination of keywords and MeSH terms, including “chimeric antigen receptor T-cells”, “CAR T-cell therapy”, “thrombosis”, “venous thromboembolism”, and “coagulation disorders”. Studies were included if they reported data on thrombotic events in patients who had undergone CAR T-cell therapy for hematological malignancies. We extracted the number of patients experiencing thrombotic events, the total number of patients in each study, and calculated the thrombotic event occurrence rate and the Wilson score interval with a 95% confidence interval.

Inclusion Criteria: We retrospectively analyzed data from 140 adult patients (age ≥ 16 years old) with r/r B-ALL, DLBCL, FL, MCL, and MM as standard of care who received CAR-T cell therapy at either Indiana University Simon Cancer Center, Indiana or the University of North Carolina Medical Center, North Carolina from July 2017 to October 2020. All patients with B-ALL received tisagenlecleucel (*n* = 3, 100%). Patients with DLBCL received either tisagenlecleucel (*n* = 21, 22.8%), lisocabtagene maraleucel (*n* = 11, 12.0%), or axicabtagene ciloleucel (*n* = 60, 65.2%). Patients with FL received axicabtagene ciloleucel (*n* = 9, 100%). All patients with MCL received brexucabtagene autoleucel (*n* = 2, 100%). Patients with MM received either idecabtagene vicleucel (*n* = 25, 73.5%) or ciltacabtagene autoleucel (*n* = 9, 26.5%). All patients received lymphodepleting (LD) chemotherapy with cyclophosphamide and fludarabine on days -5 through -3, followed by CAR T-cell infusion performed on day 0. No patients were excluded based on disease progression or receiving other disease-directed therapies after CAR T-cell infusion. 

Baseline Patient Characteristics: Laboratory values and clinical events were acquired through comprehensive review of patients’ EMR data from day -5 to day 365. BMI, past medical history, and baseline laboratory values were obtained from the day preceding CAR T-cell infusion (day 0). Patient information, including diagnosis, prior history of bleeding, prior history of vascular condition, prior history of neurological condition, prior lines of therapy, prior transplantation status, bridging therapy status, and pre-CAR T cell infusion anticoagulation prophylaxis, was acquired through review of patient EMR clinical notes. 

Relevant Coagulation Parameters: Platelet count nadir, coagulation parameters including prothrombin time (PT) peak, activated partial thromboplastin time (aPTT) peak, D-dimer peak, and ferritin peak were defined as either reported lowest or highest laboratory values in patient EMRs occurring within 30 days post-CAR T-cell infusion. Prothrombin time (PT) prolongation and activated partial thromboplastin time (aPTT) prolongation were defined as PT peak and aPTT peak exceeding either 12.5 s and 35 s with ≥3 s and 5 s, respectively.

Adverse event and toxicity assessment: Thrombotic events were included in the analysis if graded ≥ 2 using Common Terminology Criteria for Adverse Events (v 6.0) or if clinical intervention was undertaken. Assessment for CAR T toxicity was monitored routinely in patients within a 30 day hospitalization period post-CAR T-cell infusion and continued in patients with later complications. CRS and ICANS were graded using the American Society for Transplantation and Cellular Therapy (ASTCT) guidelines [26]. Management of CRS and ICANS was carried out according to institutional guidelines. The International Society on Thrombosis and Haemostasis’ (ISTH) disseminated intravascular coagulation (DIC) scoring system was used to diagnose DIC while the Padua prediction scoring system was used predict DVT risk in each patient. ISTH DIC score includes decreased platelet count (>100 = 0, 50–100 = 1, and <50 = 2), PT prolongation (<3 s = 0, <3 s to <6 s = 1, ≥6 s = 2), elevated D-dimer (0, no increase, <5 times ULN, ≥5 times ULN = 3), and decreased fibrinogen (>100 μg/mL = 0, ≤100 μg/mL = 1) with a score ≥ 5 being suggestive of overt DIC. The Padua prediction score includes status of cancer (+3), reduced mobility (+3), known thrombophilic condition (+3), recent (within 1 month) trauma and/or surgery (+2), age ≥ 70 years (+1), heart and/or respiratory failure (+1), acute myocardial infarction and/or ischemic stroke (+1), acute infection and/or rheumatological disorder (+1), obesity BMI ≥ 30 (+1), and ongoing hormonal treatment (+1) with a score of 4 indicating high risk. 

Statistical analysis: Univariate analysis was performed using the Fisher’s exact test and two-sided Mann–Whitney U test to describe associations between categorical variables and continuous variables, respectively. *p* values < 0.05 were considered statistically significant. Statistical analyses and plots were generated using JPM Pro (v 16) and R (v 4.2.1). 

## Figures and Tables

**Table 1 ijms-24-08349-t001:** Characteristics of thrombotic patients post-CAR T-cell therapy ^1^.

Patient Number	Age (Years)	Sex	Diagnosis	CAR T-Cell Product	BMI	Baseline Serum LDH (U/L)	Baseline PLT (1000 cells/μL)	PLT Nadir (1000 cells/μL)	Anticoagulant Prophylaxis?	Time to First Thrombotic Event	Type of Thrombotic Event	Treatment of Thrombotic Event	Maximum ICANS Grade	Maximum CRS Grade	Fibrinogen Nadir (mg/dL)
1	64	M	DLBCL	Axicabtagene ciloleucel	29.7	275	181	23	Yes	3	NSTEMI	Heparin, DAPT	2	2	126
2	56	M	DLBCL	Axicabtagene ciloleucel	21.4	1000	135	7	Yes	5	DVT	Rivaroxaban	3	2	51
3	54	M	DLBCL	Axicabtagene ciloleucel	31.4	536	276	9	No	15	CVST	Heparin, Rivaroxaban	4	3	57
4	52	M	DLBCL	Axicabtagene ciloleucel	36.6	355	393	203	No	30	PE	LMWH	1	2	161
5	60	F	DLBCL	Axicabtagene ciloleucel	41.2	231	79	18	No	41	PE	Apixaban	3	3	50
6	62	F	DLBCL	Axicabtagene ciloleucel	35.9	227	106	52	Yes	113	DVT	Rivaroxaban	2	2	108
7	51	F	DLBCL	Axicabtagene ciloleucel	33.3	554	294	8	Yes	346	PE	LMWH	3	2	89
8	29	M	DLBCL	Axicabtagene ciloleucel	23.4	754	270	190	No	46	PE	Heparin	0	0	350
9	69	M	FL	Axicabtagene ciloleucel	21.4	813	237	124	No	17	DVT	Apixaban	0	0	675
10	58	F	MM	Ciltacabtagene autoleucel	31.9	207	251	229	No	10	DVT	Apixaban	0	1	1769
**Average**	**55.5**	**-**	**-**	**-**	**30.6**	**495.2**	**222**	**86**	**-**	**63**	**-**	**-**	**1.8**	**1.7**	**344**

^1^ BMI measured pre-induction, baseline serum LDH and serum PLT levels measured pre-CAR T infusion, NSTEMI = non-ST elevation myocardial infarction, DVT = deep vein thrombosis, PE = pulmonary embolism, CVST = cerebral vein sinus thrombosis, DAPT = dual antiplatelet therapy, LMWH = low molecular weight heparin.

**Table 2 ijms-24-08349-t002:** Baseline patient characteristics prior to CAR T-cell infusion and their association with thrombosis ^1^.

			Thrombosis		
Patient Baseline Characteristics		All Patients	Yes (*n* = 10)	No (*n* = 130)	*p*
**Age, y, median (range)**		60 (17–81)	60 (29–69)	60 (17–81)	0.24
**Sex, *n* (%)**		-	-	-	>0.99
	Male	85	6 (60)	79 (60.8)	-
	Female	55	4 (40)	51 (39.2)	-
**Diagnosis, *n* (%)**		-	-	-	0.54
	B-ALL	3	0 (0)	3 (2.3)	-
	DLBCL	92	8 (80)	84 (64.6)	-
	FL	9	1 (10)	8 (6.2)	-
	MCL	2	0 (0)	2 (1.5)	-
	MM	34	1 (10)	33 (25.4)	-
**BMI, kg/m^2^, median (range)**		27 (17.6–53.7)	27.1 (21.4–41.2)	27 (17.6–53.7)	0.31
**Elevated baseline LDH**		-	6	35	0.06
	B-ALL	0	-	0	-
	DLBCL	30	5	25	-
	FL	5	1	4	-
	MCL	0	-	0	-
	MM	6	0	6	-
**History of bleeding, *n* (%)**		2	0	2	>0.99
**History of venous thromboembolism, *n* (%)**		23	3	20	0.21
**Pre-existing cardiovascular condition, *n* (%)**		31	2	29	>0.99
**Prior lines of therapy, *n* median (range)**		3 (1–15)	3 (2–7)	3 (1–15)	0.60
**Prior autograft transplant, *n***		50	4	46	
	B-ALL	0	0	0	
	DLBCL	18	3	15	
	FL	2	0	2	
	MCL	0	0	0	
	MM	30	1	29	
**Bridging therapy, *n* (%)**		48	4 (40)	45 (34.6)	0.74
**Preceding anticoagulation treatment, *n* (%)**		18	3 (30)	15 (11.5)	0.12

^1^ N = 140. Elevated baseline LDH defined ≥ 270 U/L. Categorical variables were analyzed by Fisher’s exact test and continuous variables by two-sided Mann–Whitney U test. Bold indicates statistically significant results. *p* values in bold are 0.05 or less.

**Table 3 ijms-24-08349-t003:** Laboratory findings within 30 days post-CAR T-cell therapy and their associations with thrombotic event ^1^.

			Thrombosis		
Laboratory Findings		All Patients (n = 140)	Yes (*n* = 10)	No (*n* = 130)	*p*
**Prothrombin time (PT), *n* (%)**		-	-	-	>0.99
	0–3 s prolongation	-	9 (90)	111 (85.4)	-
	>3 s prolongation	-	1 (10)	19 (14.6)	-
**Activated Partial Thromboplastin Time (aPTT), *n* (%)**		-	-	-	>0.99
	0–5 s prolongation	-	8 (80)	102 (78.5)	-
	>5 s prolongation	-	2 (20)	28 (21.5)	-
**D-dimer elevation**		-	-	-	
	0–3 × ULN	-	1 (10)	44 (36.7)	-
	>3 × ULN	-	9 (90)	76 (63.3)	-
**D-dimer peak, μg/mL DDU, median (range)**		1.27 (0.15–32.65)	2.917 (0.24–9.77)	1.16 (0.15–32.65)	**0.03**
**Baseline platelet count, 1000/μL, median (range)**		179.5 (19–437)	244 (79–393)	177.5 (19–437)	0.16
**Platelet nadir, ×1000/μL, median (range)**		63 (5–275)	37.5 (7–229)	63.5 (5–275)	0.69
**Ferritin peak, ng/mL, median (range)**		660 (15.4–18784)	776.6 (38.1–12,204)	660 (15.4–18,784)	0.99
**Fibrinogen nadir, mg/dL, median (range)**		227 (50–1769)	117 (50–1769)	248.5 (61–980)	0.08
**ISTH DIC score, *n* (%)**		-	-	-	0.48
	Score <5	-	6 (60)	93 (71.5)	-
	Score ≥ 5	-	4 (40)	37 (28.5)	-
**Padua Prediction Score**		-	-	-	0.75
	Low Risk (Score < 4)	-	6 (60)	70 (53.8)	-
	High Risk (Score ≥ 4)	-	4 (40)	60 (46.2)	-
**ICANS grade, *n* (%)**		-	-	-	**0.04**
	Grade 0	-	3 (30)	86 (66.2)	-
	Grade ≥ 1	-	7 (70)	44 (33.8)	-
**CRS grade, *n* (%)**		-	-	-	>0.99
	Grade 0	-	2 (20)	29 (22.3)	-
	Grade ≥ 1	-	8 (80)	101 (77.7)	-

^1^ Frequency of treatment-emergent abnormal laboratory values and median peak values in patients with thrombosis, and ICANS and CRS grade 0 vs. grades ≥ 1. Total evaluable represents the total number (for categorical variables) or median value (for continuous variables) from patients with available data for analysis. Normal range for D-dimer was < 0.25 μg/mL D-dimer units (DDU). ISTH DIC score includes platelets (>100 = 0, 50–100 = 1, and <50 = 2), PT prolongation (<3 s = 0, <3 s to <6 s = 1, ≥6 s = 2), D-dimer (0, no increase, <5 times ULN, ≥5 times ULN = 3), and fibrinogen (>100 μg/mL = 0, ≤100 μg/mL = 1), and a score ≥ 5 is suggestive of overt DIC. Padua Prediction score includes status of cancer (+3), reduced mobility (+3), known thrombophilia condition (+3), recent (within 1 month) trauma and/or surgery (+2), age ≥ 70 years (+1), heart and/or respiratory failure (+1), acute myocardial infarction and/or ischemic stroke (+1), acute infection and/or rheumatological disorder (+1), obesity BMI ≥ 30 (+1), and ongoing hormonal treatment (+1) with a score of 4 indicating high risk of DVT. Laboratory findings were fully evaluable (*n* = 140) except for D-dimer elevation (*n* = 130). *p* for categorical variables was calculated by Fisher’s exact test and for continuous variables by two-sided Mann–Whitney U test. *p* values in bold are 0.05 or less.

## Data Availability

The data that support the findings of this study are available on request from the corresponding author. Restrictions apply to the availability of these data, which were used under license for this study. Data are not publicly available due to privacy or ethical restrictions, but may be available from the authors upon reasonable request and with permission from the relevant institutional review boards.

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
