# Peer review of "Thrombotic Events Are Unusual Toxicities of Chimeric Antigen Receptor T-Cell Therapies"

_ijms, 2023, doi:10.3390/ijms24098349_

Round 1

Reviewer 1 Report

This article reports on the incidence of thrombotic events after CAR T cell therapy at two institutions and details the 10 identified cases. As there are currently only a few reports on the incidence and phenotype of thrombotic complications post-CAR T cell therapy, this work provides a meaningful addition to the literature. Because the overall incidence of thrombotic events is low, the potential to identify predictive risk factors is limited.  The analysis of CRS and neurotoxicity severity referenced in text is not consistent with the data included in tables, as detailed below.

Introduction

76-77 The meta-analysis of these cases is atypical in the introduction, if presenting as a mini systematic review could incorporate in methods/results.

Results

88 Presence of thrombotic events in a wide range after CAR T cell therapy can not be automatically attributed as “related.” The term “associated” may be more appropriate

90 Please indicate whether any of the DVTs were line-associated

144-145 The text indicates that severity of ICANS and CRS were considered, but this is inconsistent with the content of Table 3 including only Grade 0 vs Grade > 1, considering these toxicities as only yes/no rather than severity. The title of Table 3 in the legend specifically states ICANS 0-2 vs 3-4 was considered, but this is not reflected in the analysis. Because of the high rate of CRS, considering severity (grade 3-4 vs none or grades 1-2) is a key analysis to include.

Discussion

183-184 References association with severity of CRS/ICANS, which as above is not actually included in the reported results.

189 Use of “likely” is strong based on the available evidence, “possible” would be more consistent

192-193 If systemic inflammation is important for VTE development, why is there an association with ICANS but not CRS?

Methods

213 Stating “we consecutively enrolled” implies a prospective analysis, please rephrase to make clear this is a retrospective analysis

225 Presence and duration of central line use may be another important baseline characteristic

227 Why was day 365 chosen as the inclusion interval? The one case occurring at day 346 seems to be an outlier from the others. Were patients excluded if they had disease progression or received other disease-directed therapies after CAR T infusion? If not, this limits ability to attribute thrombotic events directly to the CAR infusion. 

Author Response

Response to Reviewer #1 CAR-T Coagulation Paper

We would like to express our gratitude for Reviewer #1's insightful comments and suggestions that have helped us to refine and improve the quality of our manuscript. We believe that the revisions made in response to their feedback have enhanced the clarity and accuracy of our findings, as well as highlighted the limitations and future research directions in this area of study. We have addresses all of the comments in a point-by-point letter in blue.

Reviewer 2 Report

The authors provide a retrospective analysis on thrombotic events after CART-cell therapy.
Analyzing 140 patients who were treated with CAR T-cell therapy for various hematologic malignancies, the incidence of thrombotic events related to CAR T was 7.14%.  

Overall, the manuscript is well written and I have only a few comments:

1.       Abstract: Recommend to include the time of onset of thrombotic events after CAR-T

2.       Introduction: It would strengthen the paper to name and reference a few more paper on rare complications after CAR-T such as Parkinsonian-like movement disorders after BCMA CAR T (O Van Oekelen, Nature Medicine 2021 or sarcoidosis like flare-up after CART (Leipold A, Leukemia 2023).

3.       Results: Line 100: the time to thrombotic event from CAR T-cells therapy varied with a range between 3 and 346 days.  Are authors sure that an event occurring 346 days is related to CAR-T? Did the authors check for persistence of CAR T-cells in the peripheral blood in this patient?

4.       Conclusion: recommend toning down a little.  From my point of view, there is no urgent need for risk stratification of CART patients for thrombotic events.  The incidence seems to be slightly higher to what is expected in this group of patients and compares well to other immunological interventions such as allogeneic stem cell transplantation. Yet, it is an important complication and deserves publication.

Author Response

Response to Reviewer #2 CAR-T Coagulation Paper

We would like to express our gratitude for Reviewer #2's insightful comments and suggestions that have helped us to refine and improve the quality of our manuscript. We believe that the revisions made in response to their feedback have enhanced the clarity and accuracy of our findings, as well as highlighted the limitations and future research directions in this area of study. We have addresses all of the comments in a point-by-point letter in blue.
